# Proposal of Treatment Strategy for Pedicle Fractures of the C2: An Analysis of 49 Cases

**DOI:** 10.3390/jcm10173987

**Published:** 2021-09-03

**Authors:** Jong-Beom Park, Sung-Kyu Kim, Hyoung-Yeon Seo, Jong-Hyun Ko, Tae-Min Hong

**Affiliations:** 1Department of Orthopaedic Surgery, College of Medicine, The Catholic University of Korea, Seoul 06591, Korea; spinepjb@gmail.com (J.-B.P.); os.office.6336@gmail.com (T.-M.H.); 2Department of Orthopaedic Surgery, Chonnam National University Medical School and Hospital, Gwangju 61469, Korea; hyseo2001@daum.net; 3Department of Orthopaedic Surgery, Chonbuk National University Hospital, Jeonju 54907, Korea; drhwata@gmail.com

**Keywords:** pedicle, pars interarticularis, fracture, C2, associated injury, characteristics, treatment

## Abstract

Spine surgeons often confuse C2 pedicle fractures (PFs) with pars interarticularis fractures. In addition, little information is available about the characteristics and treatment strategies for C2 PFs. We sought to investigate the characteristics of C2 PFs and to propose an appropriate treatment strategy. A total of forty-nine patients with C2 PFs were included in this study. We divided these patients into unilateral and bilateral C2 PF groups. The incidence rates and characteristics of other associated C2 and C2-3 injuries, and other cervical injuries, were evaluated. In addition, treatment methods and outcomes were analyzed. Twenty-two patients had unilateral C2 PFs and twenty-seven patients had bilateral C2 PFs. Among the cases of unilateral C2 PFs, all patients had one or more other C2 fractures, and twenty patients (90.9%) had one or two C2 body fractures. Meanwhile, among the cases of bilateral C2 PF, all patients had two or more other C2 fractures and one or two C2 body fractures. In unilateral C2 PFs, three patients with C2-3 anterior slip or adjacent cervical spine (C1-3) injury underwent surgery and nineteen patients (86.4%) were treated with conservative methods. In bilateral C2 PFs, three patients with C2-3 anterior slip or SCI at C2-3 underwent surgery and twenty-four patients (88.9%) were treated with conservative methods. Our results showed that C2 PFs do not occur alone and are always accompanied by other associated C2 injuries. C2 PFs should, generally, be thought of as a more complex fracture type than hangman’s fracture or dens fracture. Despite the complex fracture characteristics, most C2 PFs can be managed with conservative treatment. However, surgical treatments should be considered if the C2 PFs are accompanied by the C2-3 anterior slip and adjacent cervical spine injury.

## 1. Introduction

C2 pedicle fracture (PF) is not an uncommon injury but has been given less attention than dens fractures or hangman’s fractures [1,2,3,4,5,6,7]. The pedicle in the lumbar and thoracic spine is an important component that helps to maintain the stability of the spine, and this is similar in the cervical spine. The anatomical structure of the C2 pedicle differs significantly from other pedicles of the same cervical spine as well as the lumbar and thoracic vertebrae. Due to its distinct anatomical features, spine surgeons often do not understand the exact location of the C2 pedicle and confuse C2 PF with pars interarticularis fracture [1,5,6,7]. To date, few clinical studies have reported on C2 PFs and the existing studies are mainly anatomical studies [2,3]. As a result, little information is available about the characteristics and treatment strategy for a C2 PF in comparison with its importance. Therefore, based on what we observed while treating C2 PFs, we hypothesized as follows. “A C2 PF always occurs with other C2 fractures, and like other types of C2 fracture, it is well treated with conservative methods.” Accordingly, we conducted this study to investigate the characteristics of the C2 PF, focusing on other associated C2 and C2-3 injuries, and herein suggest an appropriate treatment strategy for a C2 PF.

## 2. Materials and Methods

This study was approved by the institutional review board at all participating sites. The design of the study was a retrospective multicenter study. The patients with a C2 PF were identified retrospectively from the database of four national trauma centers of tertiary university hospitals between January 2000 and December 2017. The inclusion criteria of this study were as follows: (i) acute trauma history; (ii) C2 PFs as diagnosed on axial, coronal, and sagittal computed tomography (CT) scans; and (iii) at least 12 months of follow-up after conservative or surgical treatments. Meanwhile, exclusion criteria included the presence of (i) previous cervical spine trauma or surgery history; (ii) pathologic fracture; and (iii) infection or inflammation disease. Among a total of sixty-five patients with C2 PFs, forty-nine patients met the inclusion and exclusion criteria. Finally, the study was conducted with forty-nine patients. Plain radiographs, CT, magnetic resonance imaging (MRI), and medical records were retrospectively reviewed. Trained research staff at each site extracted relevant data from patient medical records, surgical charts, radiological images, and other source documents. The data were transcribed into study-specific paper case-report forms with radiological images. We defined the anatomical structure according to the study of Ebraheim et al. [3]; the pedicle of the C2 vertebra is the portion posterolateral to the vertebral body, beneath the superior facet and anteromedial to the transverse foramen, while the pars interarticularis or isthmus is the narrower portion between the superior and inferior facets and located posteromedial to the transverse foramen (Figure 1). All available data concerning demographics, C2 PF pattern, neurologic status, union status, and perioperative or postoperative complications were extracted. The incidence rates and characteristics of other associated C2 and C2-3 injuries and other cervical injuries were evaluated. In addition, treatment methods and outcomes were analyzed. The adopted treatment was chosen according to the surgeon’s discretion on a patient-by-patient basis without study-defined selection criteria.

To determine the fracture types and patterns, two experienced spine surgeons were blinded to the identity of the cases. Any disagreement of fracture distinguishment between them was resolved by discussion, with input from a third surgeon (senior author); such disagreements occurred in eight cases (16.3%) and all were resolved. During follow-up, postoperative complications, treatment methods, and union status were checked.

Data were analyzed using the SPSS version 25.0 for Windows software program (IBM Corp., Armonk, NY, USA) and are presented as mean ± standard deviation. The chi-squared test and Fisher’s exact test were used for statistical analyses. A P-value of less than 0.05 was considered to be significant.

## 3. Results

### 3.1. Demographic Data

Demographic data and information regarding C2 PF type, injury mechanism, neurologic status, and other C2 fractures are summarized in Table 1. Thirty-two patients were male and seventeen patients were female. The mean follow-up period was 12.6 months. Approximately 90% of the causes of fracture were motor-vehicle accidents and falling down. Among forty-nine patients with C2 PF, twenty-two patients (44.9%) had a unilateral C2 PF and twenty-seven patients (55.1%) had a bilateral C2 PF. C2 PFs occurred in various forms; however, there was no case of C2 PFs alone and all cases (100%) had other associated C2 injuries, which are summarized in Table 2.

### 3.2. Unilateral Pedicle Fracture

In the unilateral C2 PF group, twenty-two patients (100%) had one or more other C2 fractures, including twenty body, thirteen pars interarticularis (PI), twenty superior articular facets (SAFs), three inferior articular facets (IAFs), thirteen with transverse foramina (TF), two laminae, and one spinous process (SP). Meanwhile, twenty patients with C2 body fractures had one or two C2 body fractures. Two patients had a C2-3 anterior slip and two patients had another cervical injury (C6-7 fracture-dislocation and C7 laminar fracture). All patients with unilateral C2 PF showed normal neurologic function.

### 3.3. Bilateral Pedicle Fracture

Among those with a bilateral C2 PF, twenty-seven patients (100%) had two or more other C2 fractures, including twenty-seven body, thirteen PI, twenty-four SAF, sixteen TF, and one SP. Meanwhile, all twenty-seven patients had one or two C2 body fractures. Five patients had a C2-3 anterior slip, one patient had a spinal cord injury (SCI) at C2-3, and six patients had another cervical injury (three C1 posterior arch fractures and three with other cervical SP fractures).

### 3.4. Treatment Outcomes

In terms of treatment methods, 87.8% (43/49) of all patients were treated conservatively and the remaining 12.2% (6/49) of patients underwent surgery. In the unilateral C2 PF group, three patients (13.6%) with a C2-3 anterior slip or adjacent cervical spine (C1-3) injury underwent surgery and nineteen patients (86.4%) were treated with conservative methods. In the bilateral C2 PF group, three patients (11.1%) with a C2-3 anterior slip or SCI at C2-3 underwent surgery (Figure 2) and twenty-four patients (88.9%) were treated with conservative methods (Table 3). Treatment success was defined as when no additional treatment was required after bone union or fusion. There were no statistical differences in treatment methods and treatment outcomes between the unilateral and bilateral C2 PF group (*p* > 0.05). Among the seven patients with C2-3 anterior slips, three with a C2-3 disc injury or adjacent cervical spine (C1-3) injury underwent an anterior cervical discectomy and fusion C2-3 and achieved solid fusion (Figure 3). Additionally, three patients without C2-3 disc injury or the adjacent cervical spine injury were treated with Halovest and achieved bone union (Figure 4). However, one patient with a bilateral PF, a C2-3 anterior slip, and a C1 posterior arch fracture developed non-union after a Philadelphia brace application (Figure 5). Nevertheless, this patient decided not to undergo surgery because there were no severe symptoms. There were no other complications related to treatment except for one instance of non-union.

## 4. Discussion

In our study, all PFs of C2 were paired with at least one or more other associated C2 fractures. Nevertheless, 87.8% (43/49) of patients received conservative treatments and all patients were well-treated except for one patient who developed non-union in the conservative treatment group. All other associated injuries, as well as the C2 PF, were also treated with conservative methods.

C2 PFs are not uncommon but have been rarely reported in the literature [1,2,3,5,6,7,8,9,10,11,12,13]. C2 fractures have historically been divided into three clinically relevant categories in the literature as follows: fractures of the odontoid process, hangman’s fractures (traumatic spondylolisthesis), and miscellaneous non-odontoid/non-hangman’s fractures or body fractures [6,14,15]. Among them, fractures of the odontoid process and hangman’s fractures are by far the most common, but miscellaneous non-odontoid/non-hangman’s fractures have also been reported with frequencies of between 5% and 25% [1,2,3,4,5,6,7]. In the past, C2 PFs have been treated collectively as axis body fractures, non-odontoid/non-hangman’s fractures, and miscellaneous fractures of the axis [8,16,17]. The diagnosis of C2 PFs on plain radiography is difficult due to their complex anatomical structure and lack of specific clinical symptoms. With the development of highly specific high-resolution imaging techniques such as CT with bone window, two- and three-dimensional CT, and MRI, miscellaneous C2 fractures can now better be classified by individual fracture features.

Even in the few reported articles that exist, there is some degree of confusion between PFs and pars interarticularis fractures [1,5,6,7]. Even experienced spinal surgeons may confuse C2 PFs with pars interarticularis fractures. Pars interarticularis fracture is a type of hangman’s fracture. The fundamental problem is that the use of the anatomic terminology of the pedicle and pars interarticularis in the C2 vertebra is confusing in most of the spine literature due to existing complex structures. Unlike the pedicle of the lumbar spine, there can be confusion in the C2 vertebra because the superior facet of C2 is more anterior to create a joint with the inferior facet of C1. When we have reviewed several papers on the C2 pedicle published so far [1,3,18,19,20,21,22], we found the most accurate study to be that by Ebraheim et al. [3], who reported that the pedicle of the C2 vertebra is the portion posterolateral to the vertebral body, beneath the superior facet and anteromedial to the transverse foramen, while the pars interarticularis or isthmus is the narrower portion between the superior and inferior facets and located posteromedial to the transverse foramen.

Just as pars interarticularis fracture has been confused with PF, it is possible that PF has been confused with body, superior articular facet, or pars interarticularis fractures. In our study, C2 PF was not rare and always occurred in combination with other fractures of the C2 cervical spine, such as vertebral body fracture or articular facet fracture. However, this can be understood by considering the anatomical structure of the C2 pedicle. Since the C2 pedicle is located under the superior articular facet and connects the pars interarticularis to the body, if the fracture line of the pedicle fracture is further expanded, then a body, superior articular facet, or pars interarticularis fracture can occur in conjunction. In addition, because the C2 pedicle is large and strong, fractures in other parts of the C2 are likely to occur in combination. In associated injuries, there was no type I or II dens fractures, and only type III dens fractures occurred in 49.0% due to the anatomical location of the C2 pedicle. Hangman’s fracture or type III dens fracture can occur alone in many cases. However, due to the above-mentioned anatomical characteristics of the C2 pedicle, a C2 pedicle fracture cannot occur alone but occurs as a more complex fracture type accompanied by other fractures. In this case, it should be expressed as a pedicle fracture with a hangman’s fracture or den’s fracture, not a hangman’s fracture or a dens fracture with a pedicle fracture and should, generally, be thought of as a more complex fracture type than a hangman’s fracture or a dens fracture. Of course, the treatment and prognosis of a pedicle fracture can be similar to that of a hangman’s fracture or a dens fracture. However, it will be a significant difference for surgeons to treat a fracture knowing the exact anatomical structure, fracture pattern, and characteristics and to treat a fracture without knowing it.

In general, hangman’s fractures, odontoid fractures (except type II), and most miscellaneous C2 fractures can be effectively managed with conservative treatment [5,6,9,11,23,24]. Most C2 PFs seem to be treated well by immobilization due to the well-vascularized large cancellous bone surface that exists along the fracture line [25]. However, studies assessing treatment results in large patient groups are rare and the published papers are also mis-researched, so there is no clearly established treatment method. The paper by Borne et al. [1], which included the most cases among the existing papers so far, focused on patients with pedicle-isthmus complex fractures and the number of pure PFs was unknown. Separately, Craig et al.’s paper was about superior articular facet fracture [12]. Although the superior articular facet anatomically involves some of the pedicles, superior articular facet fracture is not a true type of PF. The majority of the rest of the papers were case reports. Therefore, to our knowledge, our study has analyzed the largest number of pure PFs.

In the context of a C2-3 anterior slip, surgery is often necessary. C2 PFs by themselves are less likely to damage the spinal cord because they are often only minimally displaced, and the space of the spinal canal is wider relative to that seen with subaxial segments. Thus, injury to the spinal cord, generally, occurs only when the C2-3 disc is damaged. Of the seven patients with a C2-3 anterior slip, three were treated well with surgery, and three were well-treated with conservative treatment, but one patient who received conservative treatment for a bilateral PF, a C2-3 anterior slip, and a C1 posterior arch fracture developed non-union. Therefore, we recommend further consideration of surgical treatment when a C2-3 anterior slip is present along with adjacent cervical spine injury or disc injury.

The primary weakness of this study is its retrospective multicenter design. Moreover, the study results may be influenced by selection bias because the treatment in each case was chosen according to the surgeon’s discretion.

This is the most significant weakness of this paper. The exact surgical indication at the time of surgery was not known. However, six patients who underwent surgery had a C2-3 anterior slip with a disc injury or had a cord injury. That is, these were patients with neurologic symptoms or instability that could not be treated spontaneously. These are indications of spine surgery that are still unchanged. It is thought that conservative treatments were chosen except for these patients. The second limitation is that it is difficult to conclude that our results are applicable to all C2 pedicle fractures, although it is the largest number studied so far. Prospective studies with a larger number of patients are considered necessary.

## 5. Conclusions

C2 PFs do not occur alone and are always accompanied by other associated C2 injuries. Despite the complex fracture characteristics, most C2 PFs can be treated with conservative methods. However, we recommend that surgery should be considered if the C2 PF is accompanied by the instability of a C2-3 anterior slip with a C2-3 disc injury and an adjacent cervical spine injury.

## Figures and Tables

**Figure 1 jcm-10-03987-f001:**
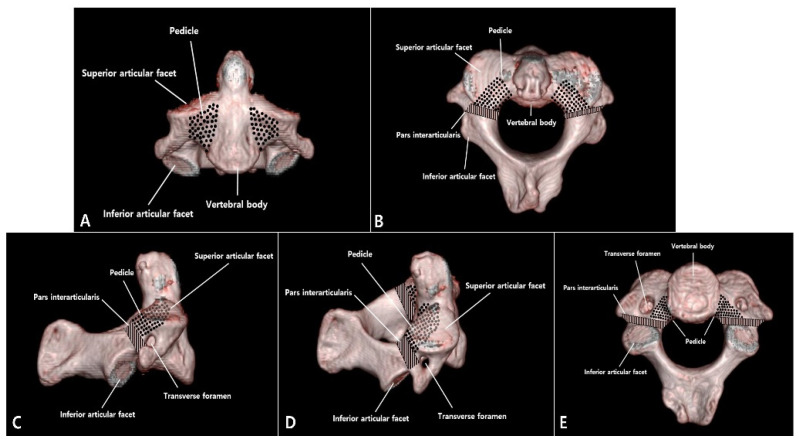
Three-dimensional computed tomography (3D-CT) scans of the C2 vertebra showing the location of the pedicle and pars interarticularis. (**A**) Anterior view. (**B**) Superior view. (**C**) Lateral view. (**D**) Oblique lateral view. (**E**) Inferior view.

**Figure 2 jcm-10-03987-f002:**
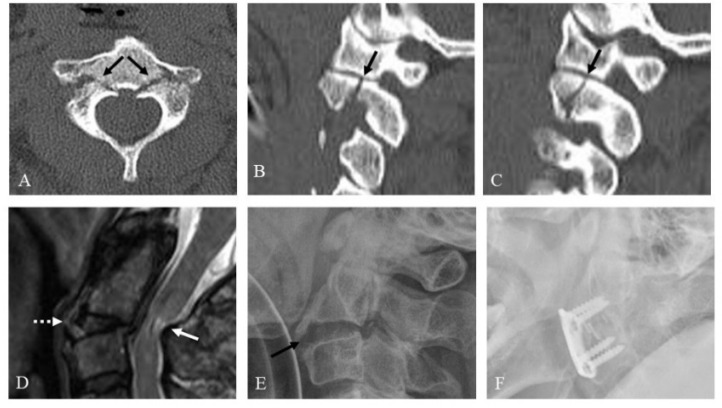
Axial computed tomography (CT) scan (**A**) showing both pedicle fractures (dark arrows). Right (**B**) and left (**C**) parasagittal CT scans showing fractures of both superior articular facets (dark arrows). Sagittal magnetic resonance imaging (**D**) showing posterior angulation of C2 on C3 (dotted white arrow) and associated spinal cord injury at C2-3 (white arrow). Pre-operative lateral radiograph (**E**) showing posterior angulation of C2 on C3 (dark arrow). At one year after surgery, follow-up lateral radiograph (**F**) indicated solid fusion of anterior cervical discectomy and fusion at C2-3.

**Figure 3 jcm-10-03987-f003:**
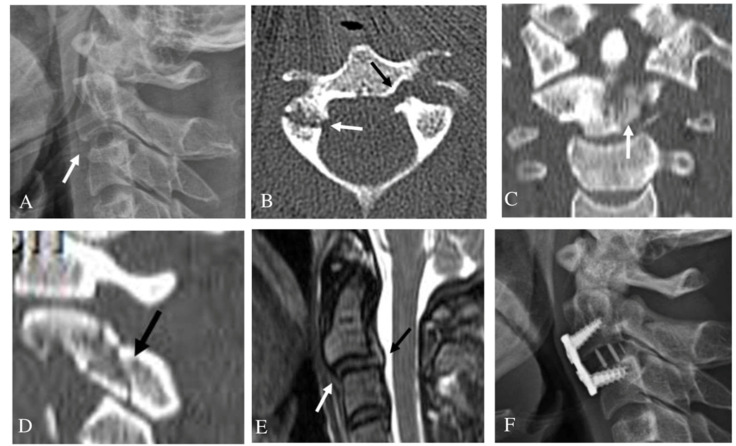
Pre-operative lateral radiograph (**A**) showing anterior slip of C2 on C3 (white arrow) and inferior articular facet fracture of C2. Axial computed tomography (CT) scan (**B**) showing left pedicle fracture (dark arrow) and right inferior articular facet fracture (white arrow). Coronal CT scans showing fracture of the C2 body (white arrow) (**C**). Right parasagittal CT scans (**D**) showing fracture of right inferior articular facet (dark arrow). Sagittal magnetic resonance imaging (**E**) showing anterior slip of C2 on C3 (white arrow) and C2-3 disc injury (dark arrow). At one year after surgery, follow-up lateral radiograph (**F**) revealed solid fusion of anterior cervical discectomy and fusion at C2-3.

**Figure 4 jcm-10-03987-f004:**
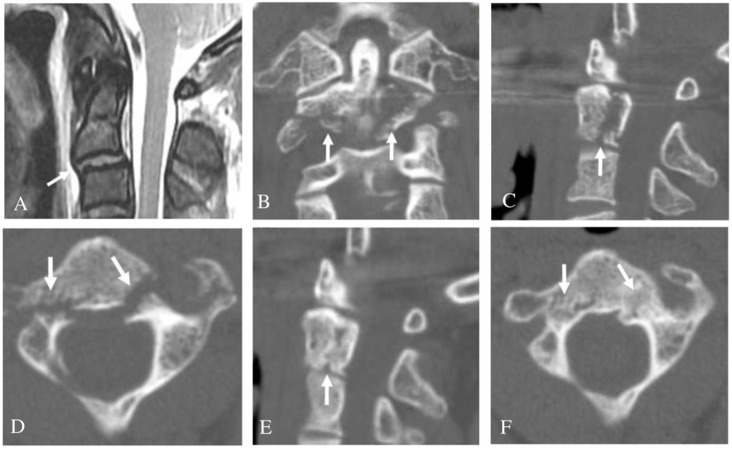
Sagittal magnetic resonance imaging (**A**) showing anterior slip of C2 on C3 (white arrow) but no evidence of C2-3 disc injury. Coronal computed tomography (CT) scan (**B**) showing comminuted fractures of the C2 body (white arrows). Left parasagittal CT scan (**C**) showing superior articular facet fracture and C2 body fracture (white arrow). Axial CT scan (**D**) showing both pedicle fractures (white arrows). At one year after Halovest application, follow-up left parasagittal CT scan (**E**) indicated bone union at superior articular facet fracture and C2 body fracture (white arrow). Follow-up axial CT scan (**F**) revealed union of both pedicle fractures (white arrows).

**Figure 5 jcm-10-03987-f005:**
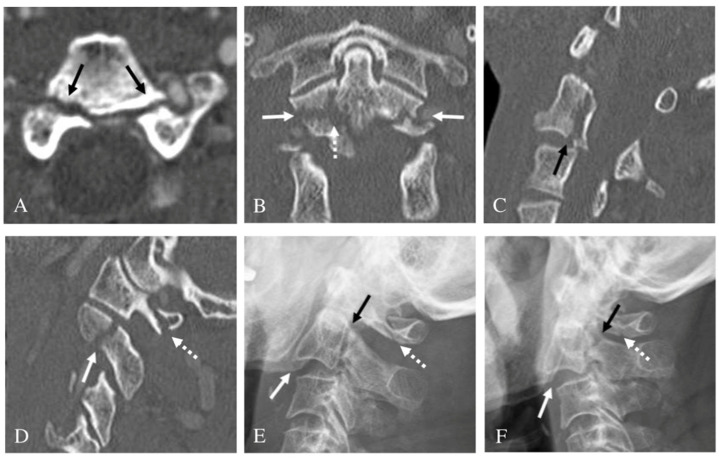
Axial computed tomography (CT) scans showing both pedicle fractures (dark arrows) (**A**). Coronal computed tomography (CT) scan (**B**) showing both pedicle fractures (white arrows) and the C2 body fracture (dotted white arrow). Sagittal CT scan (**C**) showing vertical fracture of the C2 body (dark arrow). Right parasagittal CT scans (**D**) showing fractures of the right superior articular facet (white arrow) and C1 posterior arch (dotted white arrow). Initial lateral radiograph (**E**) showing slight anterior slip of C2 on C3 (white arrow) and linear fracture line of pedicles (dark arrow) and C1 posterior arch (dotted white arrow). At one year after Philadelphia brace application, follow-up lateral radiograph (**F**) revealed progression of anterior slip of C2 on C3 (white arrow) and nonunion at fracture sites of pedicles (dark arrow) and C1 posterior arch (dotted white arrow).

**Table 1 jcm-10-03987-t001:** Demographic data of 49 cases with a C2 pedicle fracture.

Age (Years)	59.1 ± 15.4 (Range: 31–86)
Sex	Male: Female = 32:17
Follow-up (months)	12.6 ± 2.6 (range: 12–26)
Fracture type	
Unilateral pedicle fracture	22 (44.9%)
Bilateral pedicle fracture	27 (55.1%)
Injury mechanism	
Motor-vehicle accident	28 (57.1%)
Falling down	16 (32.7%)
Slipping down	3 (6.1%)
Direct external force	2 (4.1%)
Neurologic status	
Normal	48 (98.0%)
Deficit	1 (2.0%)
Other associated C2 fractures	
Yes	49 (100%)
No	0 (0%)

**Table 2 jcm-10-03987-t002:** Associated injuries of C2 pedicle fractures.

	Unilateral Pedicle Fx. (*N* = 22)	Bilateral Pedicle Fx. (*N* = 27)	*p*
Other associated C2 injuries	22 (100%)	27 (100%)	
Pars interarticularis Fx.	13	13	
Superior articular facet Fx.	20	24	
Inferior articular facet Fx.	3	0	
Transverse foramen Fx.	13	16	
Lamina Fx.	2	0	
Spinous process Fx.	1	1	
Body Fx.	20 (90.9%)	27 (100%)	0.196
*Dens Fx. (type III)*	11	13	
*Vertical Fx.*	5	14	
*Transverse Fx.*	0	1	
*Comminuted Fx.*	2	0	
*Teardrop Fx.* *(Anterior: Posterior)*	4 (1:3)	2 (0:2)	
**Anterior slip C2-3**	2	5	0.303
**Spinal cord injury at C2-3**	0	1	1.000
**Other cervical injury**	2	6	0.269

Fx. = fracture.

**Table 3 jcm-10-03987-t003:** Treatment outcome of C2 pedicle fractures.

	Total (*N* = 49)	Unilateral Pedicle Fx. (*N* = 22)	Bilateral Pedicle Fx. (*N* = 27)	*p*
**Conservative treatment**	43 (87.8%)	19 (86.4%)	24 (88.9%)	0.362
Halovest	33	14	19	
Philadelphia brace	10	5	5 (1 non-union)	
**Surgery**	6 (12.2%)	3 (13.6%)	3 (11.1%)	0.500
ACDF C2-3	4	2	2	
Posterior fusion C1-2	1	1	0	
Dens screw fixation	1	0	1	
**Success rate of treatment**	48/49 (98.0%)	100%	96.3% (1 non-union)	0.449

Fx. = fracture; ACDF = anterior cervical discectomy and fusion.

## Data Availability

All data presented in this study are available on demand from the corresponding author.

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
