# Peer review of "Proposal of Treatment Strategy for Pedicle Fractures of the C2: An Analysis of 49 Cases"

_jcm, 2021, doi:10.3390/jcm10173987_

Round 1
Reviewer 1 Report
this is a nicely conducted study, the main flaws are the short follow-up time and small sample, but given the rarity of this type of injury, this is somehow acceptable. Main issues regarding the outcomes assessment. please find specific comments below.
line 46, please report the design of this study
line 43, please report aims and hypothesis of this study
line 76, is there a reference for this classification? are there data about its reliability? if not cloud you provide data or its reliability?
line 76, please be clear about how did you assess outcomes, outcomes used at follow-up are not reported
please avoid repetition in the results section, data are well reported in tables, no need for repetition in the text
line 138-140, what you mean by outcomes? how were success or failure defined? please report
start discussion section with your main findings, please be more concise and focused on your results
Reviewer 2 Report
interesting topic, makes a specific pathology more transparent
Round 2
Reviewer 1 Report
Comments were addressed adequately
thanks for your time and effort